# Transcriptomic changes behind *Sparus aurata* hepatic response to different aquaculture challenges: An RNA-seq study and multiomics integration

Cláudia Raposo de Magalhães[1,2], Kenneth Sandoval[3], Ferenc Kagan[4], Grace McCormack[3], Denise Schrama[1,2], Raquel Carrilho[1,2], Ana Paula Farinha[1,2,5], Marco Cerqueira[1,2], Pedro M. Rodrigues[1,2]*

1 Centre of Marine Sciences (CCMAR), Universidade do Algarve, Campus de Gambelas, Faro, Portugal, 2 Universidade do Algarve, Campus de Gambelas, Faro, Portugal, 3 Molecular Evolution and Systematics Laboratory, Zoology, Ryan Institute & School of Natural Sciences, University of Galway, Galway, Ireland, 4 University of Bergen, Bergen, Norway, 5 Escola Superior Agrária de Santarém, Santarém, Portugal

* pmrodrig@ualg.pt

**Data Availability Statement:** RNA-seq data for this article have been deposited on the ArrayExpress

## Abstract

Gilthead seabream (*Sparus aurata*) is an important species in Mediterranean aquaculture. Rapid intensification of its production and sub-optimal husbandry practices can cause stress, impairing overall fish performance and raising issues related to sustainability, animal welfare, and food safety. The advent of next-generation sequencing technologies has greatly revolutionized the study of fish stress biology, allowing a deeper understanding of the molecular stress responses. Here, we characterized for the first time, using RNA-seq, the different hepatic transcriptome responses of gilthead seabream to common aquaculture challenges, namely overcrowding, net handling, and hypoxia, further integrating them with the liver proteome and metabolome responses. After reference-guided transcriptome assembly, annotation, and differential gene expression analysis, 7, 343, and 654 genes were differentially expressed (adjusted p-value < 0.01, log2|fold-change| >1) in the fish from the overcrowding, net handling, and hypoxia challenged groups, respectively. Gene set enrichment analysis (FDR < 0.05) suggested a scenario of challenge-specific responses, that is, net handling induced ribosomal assembly stress, whereas hypoxia induced DNA replication stress in gilthead seabream hepatocytes, consistent with proteomics and metabolomics' results. However, both responses converged upon the downregulation of insulin growth factor signalling and induction of endoplasmic reticulum stress. These results demonstrate the high phenotypic plasticity of this species and its differential responses to distinct challenging environments at the transcriptomic level. Furthermore, it provides significant resources for characterizing and identifying potentially novel genes that are important for gilthead seabream resilience and aquaculture production efficiency with regard to fish welfare.

database (http://www.ebi.ac.uk/arrayexpress) under accession number E-MTAB-12842. RNA-seq data is available for reviewers through this link: https://www.ebi.ac.uk/biostudies/arrayexpress/studies/E-MTAB-12842?key=4bacbe1e-b1ca-4a99-aaad-e31e825a811a.

**Funding:** Cláudia Raposo de Magalhães acknowledges a FCT PhD scholarship, Refª SFRH/BD/138884/2018. Denise Schrama acknowledges a FCT PhD scholarship, Refª SFRH/BD/136319/2018. Raquel Carrilho acknowledges a FCT PhD scholarship, Refª 2021.06786.BD. Ana Paula Farinha acknowledges postdoctoral fellowship, Refª MAR-02.05.01-FEAMP-0012 (MAR2020). Marco Cerqueira acknowledges a FCT contract, Refª 2020.02937.CEECIND.

**Competing interests:** The authors have declared that no competing interests exist.

## Introduction

Within genomics and transcriptomics, the advent of next-generation sequencing (NGS) has greatly revolutionized the study of biological systems, allowing for the rapid sequencing of whole genomes, transcriptomes, and molecular markers (e.g., single nucleotide polymorphisms (SNPs)), including those in aquatic model systems. At the time of this writing, 248 representative fish genome assemblies at the chromosome level were available in the NCBI genome database (https://www.ncbi.nlm.nih.gov/genome/), with 94% of those released only in the last four years, including the gilthead seabream (*Sparus aurata*) genome. NGS-based RNA sequencing (RNA-seq) has recently become more accessible; however, in fish, transcriptomics is still in the nascent stage. However, besides mapping and annotating fish transcriptomes, it has already offered valuable insights into many biological processes in commercially important fish species and has helped scientists tackle many challenges in aquaculture [1,2]. Using solutions such as RNA-seq to integrate the use of various optimization production criteria is pivotal for the sector's sustainability and competitiveness.

Ensuring the sustainable growth and development of aquaculture in response to its evident intensification is at the forefront of priorities for meeting the increasing fish consumption rate of the global population. In fact, aquaculture is currently the most important industry worldwide to compensate for the declining and rapidly accelerating depletion of wild fish stocks. Its production is projected to continue to increase and reach 106 million tons of aquatic animals by 2030, compared to 87.5 million tonnes registered in 2020 [3]. However, the continued increase in the number of aquatic animals produced poses many challenges for meeting the global demand for fish. Disregarding the overall farming conditions may significantly impact different measures of fish performance, and consequently, productivity [4]. Sub-optimal husbandry conditions, such as high rearing densities, can be stressful for some fish species and consequently affect growth rates, trigger aggressive/unwanted behaviours, and reduce disease resistance [5]. Furthermore, prolonged exposure to high stocking densities has been shown to negatively affect the response to subsequent stimuli such as acute net confinement [6]. Hypoxia is often associated with overcrowding and is known to induce significant physiological changes such as reduced appetite, depressed metabolic rates and muscle oxidative capacity, and a switch in substrate preference towards more oxygen ($O_2$)-efficient fuels [7]. Unpredictable physical stressors such as handling are common procedures in aquaculture farms that can increase the chances of abrasion, wounds, and infections, thereby causing severe stress [5].

Fish display different coping mechanisms to deal with environmental challenges through adaptive neuroendocrine and metabolic adjustments, collectively termed stress responses [8]. The hypothalamic-pituitary-interrenal (HPI) axis mediates this response, promoting the synthesis of glucocorticoid hormones (e.g., cortisol) that activate distinct signalling and metabolic pathways responsible for the overall physiological rearrangement needed to adapt to the new internal disturbance [9]. The liver is the leading organ in this response, managing substrate administration by synthesizing glucose and regulating somatic growth, immune response, detoxification, and synthesis of stress-related proteins [10].

High-throughput transcriptomic studies with different fish species have mainly focused on the immunological responses to pathogens and parasites [11], and on the effects of alkalinity [12], rearing density [13,14], temperature [15–18], salinity [19], ammonia [20], and fasting [21,22]. However, studies on the transcriptional effects of other aquaculture stressors are still lacking. Stress-related RNA-seq studies on gilthead seabream have focused on the effects of ultraviolet B radiation exposure in the skin [23], gill tissue response to an ectoparasite [24], whole-brain analysis of food-deprived individuals [25], and the effects of mild hypoxia in the muscle [7]. To our knowledge, no study has addressed and compared the hepatic

transcriptome response to different aquaculture challenges in gilthead seabream, a highly consumed and produced fish in the Mediterranean region [26].

Therefore, in this study, RNA-seq was employed to characterize the transcriptional machinery behind stress adaptation, underlining and quantifying the genes and gene families expressed in the liver of gilthead seabream adults in response to different aquaculture challenges, namely, overcrowding, net handling, and hypoxia. Multiomics integration was further performed to compare the most significant dysregulated biological functions in the proteome, metabolome, and transcriptome. This multi-level characterization of stress adaptation mechanisms in gilthead seabream provides valuable knowledge for the future selective breeding of more resilient commercial species that can thrive under changing conditions and adapt well to life in captivity while ensuring high welfare standards.

## Materials and methods

### Fish husbandry and ethics

Gilthead seabream (*Sparus aurata*) adults, supplied by the company "Maresa, Mariscos de Estero S.A." (Huelva, Spain) were maintained at the Ramalhete Research Station of the CCMAR facilities (Faro, Portugal) under standard rearing conditions. Throughout the experimental trials, fish were maintained in 500 L fiberglass tanks with a flow-through system with seawater from the local Ria Formosa (natural photoperiod, water temperature: $13.4 \pm 2.2°C$, dissolved oxygen level: $> 5$ mg $L^{-1}$, and salinity: $34.7 \pm 0.8$ psu). Fish were fed once daily by hand (% body weight day$^{-1}$ adjusted when necessary), with commercial feed (Standard Orange 6) from "AquaSoja, Sorgal, S.A" (Ovar, Portugal), according to the nutritional requirements of the species.

The present study was officially approved by the Responsible Body for Animal Welfare (ORBEA) of CCMAR and the Portuguese National Authority for Animal Health (DGAV) on August 26, 2019. The animal experiments followed the European guidelines on the protection of animals used for scientific purposes (Directive 2010/63/EU) and the Portuguese legislation for the use of laboratory animals, under a "Group-1" license (permit number 0420/000/000-n.99–09/11/2009) from the Veterinary Medicine Directorate, the competent Portuguese authority for the protection of animals, Ministry of Agriculture, Rural Development and Fisheries, following the category C FELASA recommendations. This manuscript adheres to the ARRIVE guidelines for the reporting of animal experiments.

### Experimental trials and sampling

Three experimental trials were conducted separately, where fish were subjected to three different challenges: overcrowding—OC, repetitive net handling coupled to air exposure—NET, and hypoxia—HYP. In each trial, fish were randomly assigned to two experimental groups: (1) the control group (CTRL) and (2) the challenged group. Each experimental group was divided into three tanks with an initial rearing density of 10 kg m$^{-3}$ (except for the high rearing density group, as described further). In the OC trial, fish with an average initial body weight (IBW) of $373.89 \pm 11.04$ g were reared under high stocking densities over 54 days. Experimental groups were established as follows: (1) CTRL– 10 kg m$^{-3}$ and (2) OC45–45 kg m$^{-3}$. In the NET trial, fish (IBW = $376.52 \pm 8.96$ g) were challenged for 45 days with nets designed for the purpose that were fitted inside the tanks: (1) CTRL–undisturbed fish (the net was equally fitted inside the tanks but not lifted) and (2) NET4 –fish were lifted and air-exposed for 1 min, four-times a week. In the HYP trial, fish (IBW = $405.74 \pm 35.14$ g) were reared under low levels of dissolved oxygen for 48 h. Experimental groups were established as follows: (1) CTRL– 100% saturated oxygen and (2) HYP15–15% saturated oxygen. Saturated oxygen levels were measured every

30 min to keep track of potential fluctuations and adjust the nitrogen injection if necessary. The zootechnical results have been previously published [27].

At the end of each trial, three fish were randomly collected from each tank and immediately anesthetized using a lethal dose (200 mg L$^{-1}$) of tricaine methanesulfonate (MS-222; Merck KGaA, Darmstadt, Germany). Liver samples were collected, chopped, immediately frozen in liquid nitrogen, and stored at -80°C until further use. According to standard aquaculture practices, fish were starved for 48 h before sampling to clean the digestive tract.

A schematic workflow of the methodology is provided in S1 Fig.

## Liver RNA sequencing

**Total RNA extraction and purification.** Total RNA was extracted from 70 mg of gilthead seabream liver samples (n = 9, 3 fish per tank; tank unit as a biological replicate) using TRI reagent® (T9424, Sigma-Aldrich, Merck), following the manufacturer's instructions, with slight modifications. Briefly, after homogenizing the tissue with an autoclaved micropestle in 1 ml TRI reagent®, homogenates were centrifuged at 12,000 × g for 10 min at 4°C and the supernatant was left to stand at room temperature (RT) for 5 min. Phase separation was achieved with 200 μL of cold chloroform (-20°C), followed by vortexing, incubation for 15 min at RT, and centrifugation at 12,000 × g for 30 min at 4°C. RNA isolation from the aqueous phase was performed using 500 μL of cold isopropanol (-20°C), followed by vortexing. For RNA precipitation, samples were allowed to stand for 1 h at -20°C followed by centrifugation at 12,000 × g for 15 min at 4°C. The pellets were then washed twice with 1 ml 75% cold EtOH (-20°C), centrifuged at 12,000 × g for 8 min at 4°C, and dried for 5–10 min, on ice in a fume hood. The pellets were resuspended in 50 μL of RNase-free water in a ThermoMixer® C (Eppendorf, Hamburg, Germany) at 55°C for 10 min at 500 rpm. RNA purification and DNase I treatment were performed using the Isolate II RNA Mini Kit (BIO-52073, Meridian BioScience®, Cincinnati, OH, USA), according to the manufacturer's instructions. The yield and purity of extracted RNA were assessed using a NanoVue Plus spectrophotometer (GE Healthcare, Chicago, IL, USA). Total RNA quality and integrity were checked using a 2200 TapeStation (Agilent Technologies, Santa Clara, CA, USA), and all samples with an RNA integrity number (RIN) > 7 were considered for sequencing.

**Library construction and RNA sequencing.** RNA-seq libraries were prepared from 1 μg of total RNA using the Illumina TruSeq™ Stranded mRNA Library Prep Kit (Illumina Inc., San Diego, CA, USA) according to the manufacturer's instructions. All RNA-seq libraries were paired-end (PE) sequenced (2 × 151 bp) with dual indexing on an Illumina NovaSeq 6000 System, with poly-A selection, according to the manufacturer's protocol (TruSeq Stranded mRNA Reference Guide # 1000000040498 v00). The sequencer generated BCL/cBCL (base call) binary files, which were then converted into FASTQ files using bcl2fastq. Raw sequenced data were deposited in the ArrayExpress [28] database (http://www.ebi.ac.uk/arrayexpress) under accession number E-MTAB-12842. Approximately two billion PE reads were obtained from the 54 sequenced samples, with an average of approximately 37 million reads per sample (S1 Table). Library construction and RNA sequencing were performed by Macrogen, Inc. (Seoul, South Korea).

**Quality assessment, reads mapping and differential gene expression analysis.** Quality control (QC) analysis of raw reads was performed using FastQC v0.11.9 (Andrews 2010). Raw data were processed using Fastp v0.22.0 [29] to remove adapters, filter-out low-quality and short reads (cut-off = 100 bp), and perform base correction in overlapped regions. Fastp was also used to calculate the Q20, Q30, GC-content, and sequence duplication levels of the clean data. Trimmed reads were inspected again using FastQC to ensure their quality and were then

used for subsequent analyses. Mapping to the *Sparus aurata* reference genome (Genome assembly: GCA_900880675.1, https://www.ensembl.org/Sparus_aurata/Info/Index) was carried out using the splice-aware STAR aligner v2.7.10 [30], with the following settings: overhang– 150 bp, length (bases) of the SA pre-indexing string– 13, minimum intron length– 20, minimum alignment score normalized to read length– 0.4, minimum matched bases normalized to read length– 0.4 and output BAM files sorted by coordinate. Mapped reads were extracted from the BAM files using SAMTools v1.9 [31] and investigated using the genome browser Integrative Genomics Viewer (IGV) v2.15.4 [32]. To improve annotation, a reference-guided transcriptome assembly was performed using Stringtie v2.1.1 [33]. Potential transcripts were assembled individually for each sample and merged to generate a non-redundant transcriptome, which was subsequently compared to the reference annotation file (GTF) using gffcompare v0.11.2 [34]. A new alignment of the reads was performed using STAR with the new GTF file and the same settings as those described above. Alignment QC was performed using Qualimap v2.2.1 [35]. All results from the previous steps were merged with MultiQC v1.13 [36] (the report is provided in S1 File). All analyses were performed using the CCMAR's high-performance computing (HPC) facility, CETA.

The number of reads per gene was counted while mapping within STAR using reverse strandedness counts. Differential expression analysis (DEA) was performed by importing the raw read counts of each sample into the R package DESeq2 v1.36.0 [37] from Bioconductor. Genes with low expression were removed, normalization was performed according to sequencing depth and RNA composition, and variance stabilizing transformation (VST) was applied for visualization. The threshold for differentially expressed genes (DEGs), calculated using Wald's test, was an adjusted p-value (Benjamini-Hochberg correction) < 0.01 and log2| fold-change| (LFC) > 1.0, after Bayesian shrinkage [38]. Principal component analysis (PCA) was achieved with the Bioconductor R package PCAtools v2.8.0 [39].

**Functional enrichment analysis.** Annotation of unknown genes and transcripts was performed using the HMMER v3.3 nhmmer tool [40] for homology search against *Danio rerio* (cut-off threshold of E-value < 0.01). Queries that matched no hits within the threshold were reanalysed with Pannzer2 [41] (cut-off threshold of positive predictive value (PPV) > 0.5) by first extracting candidate open reading frames (ORFs) of at least 70 amino acids and predicting potential peptides using TransDecoder v5.7.0 (https://github.com/TransDecoder/TransDecoder).

Prior to enrichment analyses, *Danio rerio* orthologs of annotated genes were searched for all identifiers using g:Profiler [42]. Gene ontology (GO), Kyoto Encyclopedia of Genes and Genomes (KEGG), and REACTOME overrepresentation analyses (ORA) were performed using enrichGO() and enrichKEGG() functions from the R package clusterProfiler v.4.4.4 [43] and enrichPathway() from the ReactomePA package v1.40.0 [44]. All terms were considered enriched with a cut-off value of < 0.05 for the adjusted p-values (Benjamini-Hochberg correction). Gene set enrichment analysis (GSEA) [45] on the aforementioned knowledgebases was performed using the clusterprofiler package. The genome-wide annotation of zebrafish from Bioconductor (R package org.Dr.eg.db v3.8.2) [46] was used for mapping in all enrichment analyses. No significantly enriched terms were found for the OC trial; therefore, it was excluded from subsequent analyses. Visualization was achieved with packages ggplot2 v.3.4.0 [47] and enrichplot v1.16.2 [48].

Multiomics integration was performed using the corresponding and previously published proteomic and metabolomic data from the same fish specimens [49]. KEGG and REACTOME ORA of proteomics datasets were likewise performed using the clusterProfiler R package, whereas for metabolomics datasets, analysis was performed using the MetaboAnalyst 5.0 Enrichment analysis web-based tool [50] and the REACTOME Analysis tools [51].

All figures were generated using the open-source graphics editor Inkscape (http://www.inkscape.org/).

## Results and discussion

### Overview of RNA-seq data and differential expression analysis

Trimming and quality filtering of raw reads resulted in an average of 2.71% discarded reads per sample, mainly due to short size (cut-off was set at 100 bp) and/or low quality. Reads that passed the filter ranged between 14.5 and 22.5 million per sample, with an average length size of 146 bp and a GC content of 49.57%. Regarding the first alignment, more than 90% of the trimmed reads were mapped to the reference genome (uniquely mapped), of which 17.85%, on average, mapped to no features (i.e., unannotated regions of the genome). Reference-guided transcriptome assembly was performed to improve genome annotation, enabling the discovery of 3,637 putative new genes and 5,036 transcripts (i.e., no overlap with any reference gene/transcript) out of a total of 31,834 assembled genes. Summary statistics of the comparison between the assembled transcriptome and the reference genome are displayed in Table 1. The new alignment with the assembled transcriptome revealed an average mapping rate of 91.79%, with 2.33% of the reads mapping to no features. The alignment QC results also showed an improvement in the genomic origin of the reads, as those mapped to exonic regions increased from an average of 64% in the first alignment to 89% after the new alignment with the assembled transcriptome. Significant homologies (E-value < 0.01) with *Danio rerio* were retrieved for 24% of the unknown genes.

Gene counts were then imported into R for differential expression analysis (DEA) and low expression genes were removed, resulting in three datasets with 17,775, 17,361 and 17,838

**Table 1. Summary statistics of gffcompare.**

| Data summary | | |
|---|---|---|
| Query mRNAs | 125523 in 32533 loci (118947 multi-exon transcripts) | |
| Reference mRNAs | 73301 in 27314 loci (70848 multi-exon) | |
| Matching intron chains | 70848 | |
| Matching transcripts | 73093 | |
| Matching loci | 27110 | |
| Missed exons | 0/378760 (0.0%) | |
| Novel exons | 44268/475793 (9.3%) | |
| Missed introns | 396/321758 (0.1%) | |
| Novel introns | 21709/373737 (5.8%) | |
| Missed loci | 0/27314 (0.0%) | |
| Novel loci | 6383/32533 (19.6%) | |
| **Accuracy estimation** | | |
| | Sensitivity | Precision |
| Base level | 100.0 | 71.3 |
| Exon level | 97.6 | 78.5 |
| Intron level | 99.7 | 85.8 |
| Intron chain level | 100.0 | 59.6 |
| Transcript level | 99.7 | 58.2 |
| Locus level | 99.3 | 79.8 |

Comparison between the experimental transcriptome assembled with Stringtie and the *Sparus aurata* reference genome.

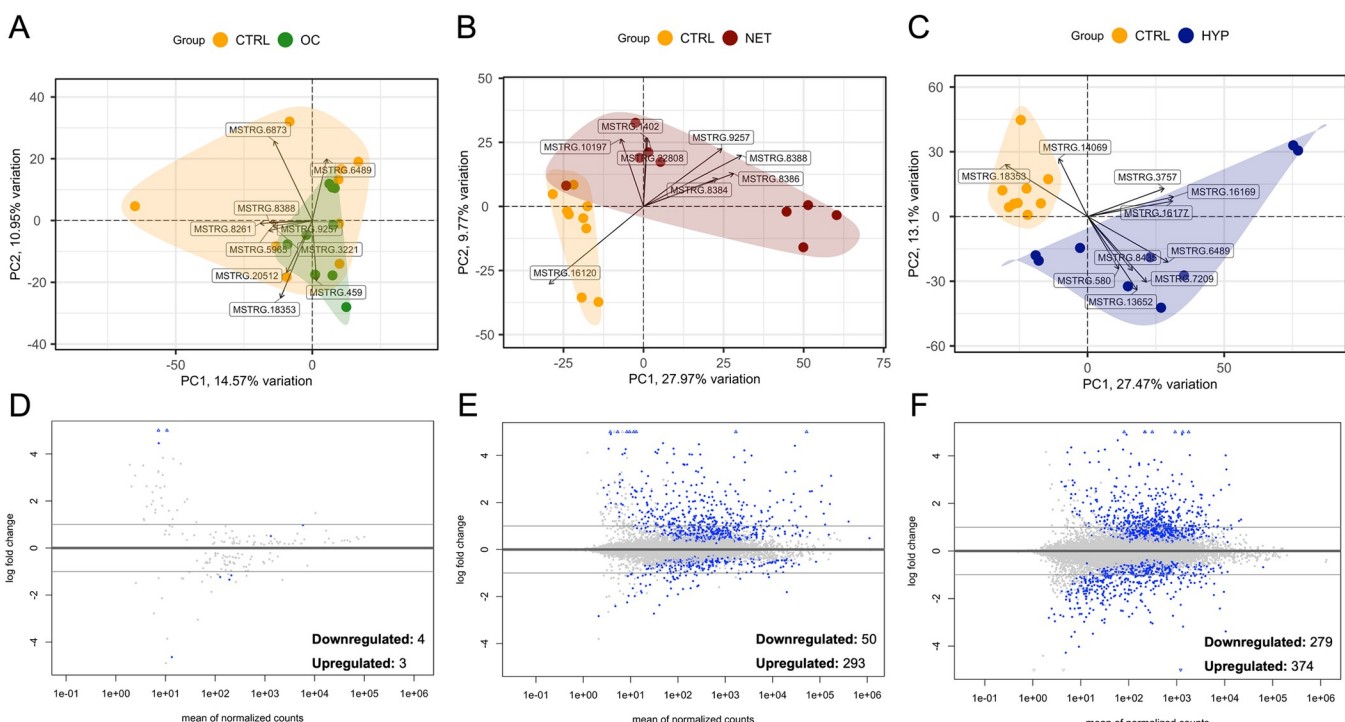

**Fig 1. Summary of the exploratory and differential analyses results of RNA-seq data.** Biplots represent the principal component analyses (PCA) of the liver transcriptome of gilthead seabream submitted to overcrowding (A), net-handling (B), and hypoxia (C). Experimental groups are distinguished by different colours, as indicated in the legend. Arrows depict the top loadings. MA plots of the shrunken LFCs indicate differentially expressed genes: (D) overcrowding, (E) net handling, (F) hypoxia. Blue points represent padj > 0.01, and horizontal lines indicate the threshold of log2|fold-change| > 1.0.

assembled genes for OC, NET and HYP trials (S2 Table). PCA biplots with VST-transformed counts (Fig 1) showed a clear separation between control and treated samples for the NET and HYP trials, along the first and the two first principal components (PC), accounting for 37.74% and 40.57% of the total data variance, respectively. Considering the PCA of NET samples, MSTRG.16120, MSTRG.8388, and MSTRG.8386, coding for pentraxin-like and hepcidin-like proteins, were the top three genes with the highest absolute loading values in PC1 (Fig 1B). Regarding HYP, the top three genes that presented the highest absolute loading values in the first PC were MSTRG.16169 and MSTRG.16177, encoding two proteins from the heat shock protein 70 family, and MSTRG.18353 encoding a protein from the cytochrome P450 family 2 (Fig 1C). In contrast, in the OC biplot, an overlap between the control and experimental group was observed (Fig 1A).

DEA retrieved 7, 343, and 654 DEGs (assembled gene IDs) (padj < 0.01, LFC >1) among the OC (Fig 1D), NET (Fig 1E) and HYP (Fig 1F) trials, respectively (S3 Table). Of these, 1, 15, and 22 genes, respectively, were not annotated. These numbers demonstrate that the overcrowding challenge had a drastically lower impact on the hepatic transcriptome than hypoxia and net handling, possibly suggesting adaptation/habituation of the animals or a higher resistance to this condition in this species. This trend was also observed for both the liver proteome and metabolome, as previously reported [49]. The low plasma cortisol levels found and previously reported also corroborate this hypothesis [27]. In this context, a recent study compared the response of European seabass and gilthead seabream to chronic overcrowding and found higher resilience of the latter in terms of plasma hormones and gene expression [52]. Additionally, a transcriptomic study with gilthead seabream juveniles subjected to food deprivation

and high stocking densities also showed that different stressors are handled by different stress pathways [53], supporting the challenge-specific responses observed here. This demonstrates the great adaptive plasticity of gilthead seabream in different farming and challenging environments. In fact, the underlying genetic basis of this trait has been recently demonstrated and attributed to high rates of gene duplication and mobile genetic elements, which might favour the acquisition of novel gene functions [54].

**Net handling induced ribosomal assembly stress coupled to downregulation of insulin growth factor signalling in gilthead seabream hepatocytes.** Gene set enrichment analysis (GSEA) based on GO biological process (BP) (Fig 2A and 2D), KEGG (Fig 2B and 2E), and REACTOME (Fig 2C and 2F) databases revealed 183 enriched terms (FDR < 0.05) for NET trial genes (S4 Table). The top significantly downregulated processes i.e., those with the lowest normalized enrichment score (NES) in all three databases, were mainly related to rRNA processing, ribosome biogenesis, and translation initiation (Fig 2A–2C). Interestingly, all of these processes were upregulated at the proteome level, as retrieved by the ORA of the proteomics dataset (S5 Table). The simultaneous upregulation of protein homeostasis genes and downregulation of ribosomal protein genes (RPGs), followed by disruption of various steps in ribosome biogenesis (rRNA production, processing, or ribosome assembly), as further explained, suggests that net handling induced ribosomal assembly stress through a response similar to the Ribosome Assembly STress Response (RASTR), previously described in yeast and humans [55,56]. This dysregulation of ribosome biogenesis and assembly can result in free ribosomal proteins (RPs) [57], which might explain their upregulation at the proteomic level [49]. The

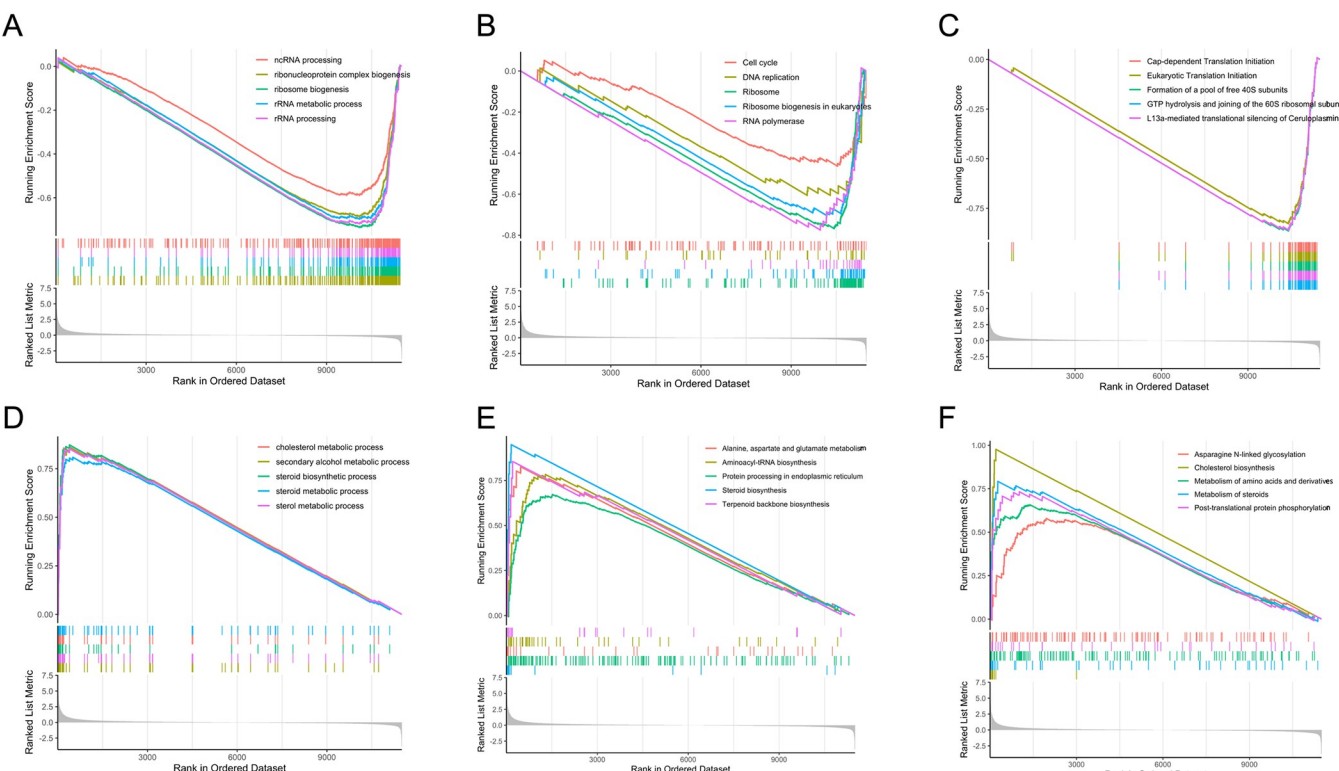

**Fig 2. GSEA of the liver RNA-seq data of gilthead seabream submitted to net handling.** Analysis was based on GO (A,D), KEGG (B,E), and REACTOME (C,F) databases, sorted by normalized enrichment score (NES) inferred from permutations of the gene set and false discovery rate (FDR). On the x-axis, the genes were ranked from the most upregulated (left end) to the most downregulated (right end). The y-axis represents the running enrichment score (ES). First line indicates downregulated pathways whereas bottom line indicates upregulated pathways.

RASTR regulatory pathway is essential for transcription regulation to maintain proteome homeostasis, thus avoiding the accumulation of defective and/or unassembled ribosomal proteins.

Ribosome assembly is a highly complex process associated with cell growth and proliferation. It monopolizes an enormous fraction of biogenic capacity and requires the coordinated work of rRNA, RPs, and other factors. In eukaryotes, ribosomes are comprised of four rRNAs (28S, 18S, 5.8S, and 5S) and 79 highly conserved RPs organized in a small (40S) and a large subunit (60S). The first three rRNAs are synthesized by RNA polymerase I (Pol I) along with other factors in the nucleolus, while 5S rRNA is transcribed separately by Pol III in the nucleoplasm. The pre-rRNAs are then assembled with RPs and exported to the cytoplasm for final maturation [58]. Unsurprisingly, this process is strictly regulated spatiotemporally through a myriad of quality control checkpoints involving a staggering number of factors [59]. Regulation at the rRNA level can occur through different signalling pathways, such as the PI3K/AKT, MAPK/ERK, and mammalian rapamycin protein kinase (mTOR) pathways [59,60]. Activation/repression of rDNA transcription by these pathways occurs through the transcriptional modulation of both Pol I and III, by interacting with specific transcription factors (TFs) [59,61]. Besides recruiting TFs, the action of the PI3K/AKT pathway on RNA polymerases is also mediated by the factor c-Myc, which is considered a major regulator of ribosome assembly [62]. Interestingly, *myca*, and the activator protein *mycbp*, were downregulated in net-handled fish (S3 Table). The significant upregulation of the pathway "AUF1 (hnRNP D0) binds and destabilizes mRNA (ID: R-DRE-450408)" observed in the proteomics data ORA (S5 Table) might corroborate the downregulation of the c-Myc gene, as the AUF1 complex binds and destabilizes mRNAs encoding, among others, c-Myc, interleukin-1 beta (IL1B), and cyclin-dependent kinase inhibitor 1 (CDKN1A). Accordingly, the latter (*cdkn1a*) was also significantly downregulated in net-handled fish (LFC = -2.83, padj = 0.006). Maf1 is also a central negative regulator of Pol III transcription. Additionally, it was shown to suppress the transcription of the TATA-binding protein (TBP), a transcription factor used by all nuclear RNA polymerases [63]. Genes *maf1* and *tbp* were found to be up- and downregulated, respectively, in net-handled fish (S2 Table), although the difference was not considered statistically significant (padj > 0.01). Furthermore, the downregulation of the pathways "RNA Polymerase III Transcription Initiation From Type 3 Promoter (ID: R-DRE-76071)", "FoxO signalling pathway (ID: dre04068)" (S4 Table), "PI3K cascade (ID: R-DRE-109704)" and "IGF1R signalling cascade (ID: R-DRE-2428924)" (S5 Table) in net-handled fish suggests a downregulation of Pol III and consequently a repression of the 5S rRNA transcription, which may lead to an inability to assemble the ribosomes properly.

Type 1 insulin-like growth factor (IGF1) is an extracellular growth factor that can activate the PI3K/AKT signalling pathway (Fig 3). In fact, *igf1* and *igf1rb*, coding for the growth factor and its receptor, respectively, were found to be downregulated in the fine flounder (*Paralichthys adspersus*) skeletal muscle after crowding stress [64], in the liver of coho salmon (*Oncorhynchus kisutch*) 16h after acute handling stress [65], and in the liver of gilthead seabream exposed to acute confinement [66]. Moreover, *igfbp1a*, coding for the IGF binding protein 1a, which binds IGF, with high affinity, in the extracellular environment, was significantly upregulated in this study (LFC = 2.46, padj = 0.003). This protein is mainly produced in the liver and prevents IGF1 from binding to its transmembrane receptor (IGF1R) and inducing cellular growth [67]. The elevation of this protein in response to stress in the liver of gilthead seabream suggests an important role in adaptation mechanisms, most likely shifting the energy from somatic growth towards stress-responsive pathways to promote survival. In rainbow trout (*Oncorhynchus mykiss*) exposed to handling and confinement stress, reduced IGF1 signalling in peripheral tissues was also observed due to the upregulation of IGFBP1 [68]. The

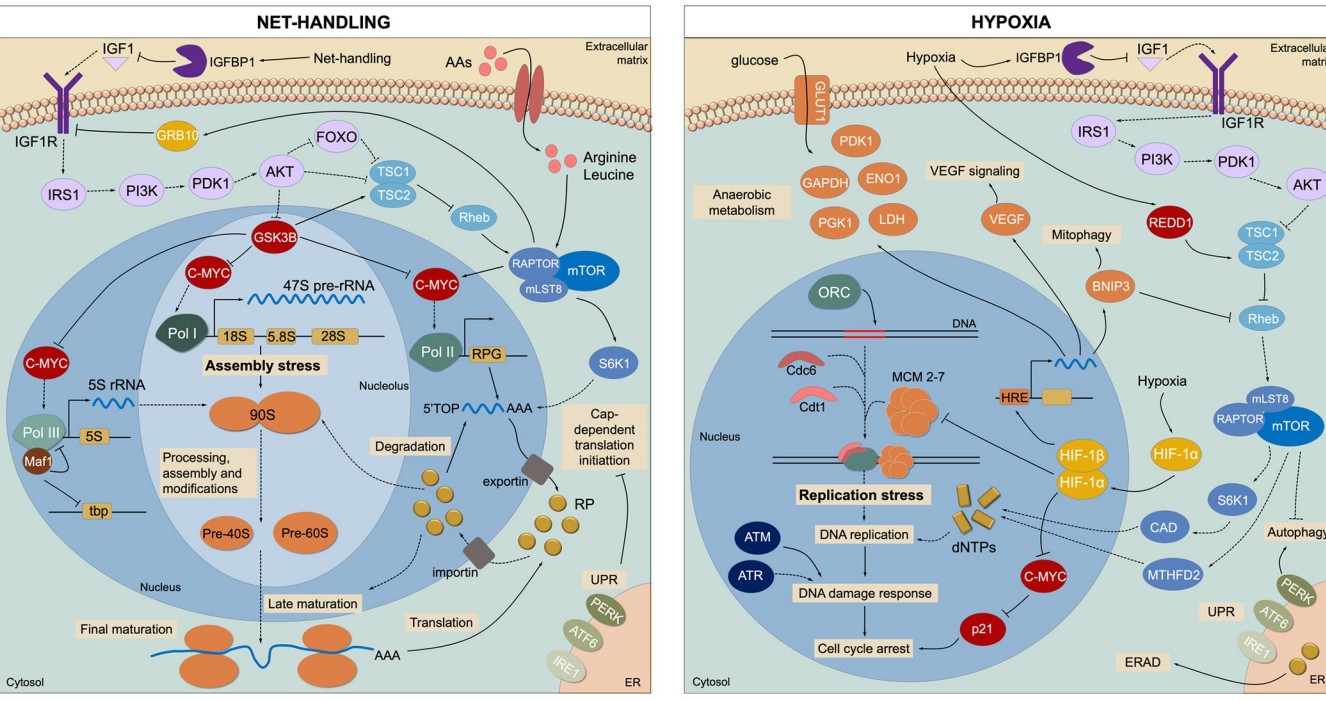

**Fig 3. Proposed stress response network in gilthead seabream hepatocytes subjected to net handling and hypoxia.** Dashed arrows indicate downregulated pathways, whereas solid arrows represent unchanged or upregulated pathways.

mTORC1 is also known to inhibit IGF1R through a negative feedback loop involving growth factor receptor-bound protein 10 (GRB10) [69]. Concomitantly, the expression of the corresponding gene (*grb10b*) was upregulated in these fish (LFC = 0.62, padj = 0.017).

RPGs are among the most highly expressed genes in most cell types, and their architecture increase the complexity of ribosome biogenesis [58]. mTOR signalling regulates RPGs' expression and promotes the synthesis of RPs in two steps. First, it induces the transcription of RPGs and small nucleolar ribonucleoproteins (snoRNPs), necessary for ribosome assembly, through Pol II. Second, by promoting the translation of RPs mRNAs through their 5' terminal oligopyrimidine (TOP) motifs, in an RPS6KB1-dependent manner [61]. Intriguingly, *rps6kb1b* is upregulated in net-handled fish (LFC = 0.82, padj = 0.008) suggesting an upregulated translation, however the GSEA indicated that "Cap-dependent Translation Initiation (ID: R-DRE-72737)" was negatively enriched (Fig 2C). This was mainly associated with the downregulated genes encoding for the different subunits of the eukaryotic initiation factor 3 (eIF3), which targets and initiates the translation of a specialized repertoire of mRNAs involved in cell proliferation. The downregulation of its transcripts may also be related to the impairment of ribosome assembly, as the 40S subunit is required with eIF3 to form the translation pre-initiation complex (PIC) [70]. In contrast, the protein levels of the six eIF3 subunits were upregulated, as previously reported [49]. Moreover, specific overexpressed RPs have been shown to autoregulate their transcripts by alternative splicing, redirecting them to degradation through different systems, such as nonsense mediated decay (NMD), ribonucleases, or exosomes [71]. PTMs are another mechanism of RPG regulation that modifies protein stability and function, with ubiquitination and phosphorylation being the two most commonly occurring processes [72]. In fact, "Post-translational protein phosphorylation (ID: R-DRE-8957275)" was one of the positively enriched pathways in net-handled fish (Fig 2F). At this step, RPs are translated in the

cytoplasm, imported into the nucleus for ribosome assembly, and then exported back into the cytoplasm for maturation. Unsurprisingly, this causes substantial demands on nuclear import and export machinery, and any perturbation at these steps can also impair ribosome biogenesis. In net-handled fish, the *ipo7* and *ipo4* genes, that encode the importin 7 and 4 import factors, were found to be downregulated (LFC = -0.39, padj = 0.006 and LFC = -0.58, padj = 0.012, respectively), along with the pathway "Nucleocytoplasmic transport (ID: GO:0006913)" (S4 Table).

Overall, these results suggest that inhibition of IGF1 by net handling stress downregulated PI3K/AKT and mTOR signalling pathways, resulting in the repression of RNA polymerase activity and consequent perturbation of the ribosome assembly process. Dysfunctional ribosomes are associated with a panoply of human disorders called ribosomopathies and are, in fact, behind several cancers [73], however this association has not yet been explored in fish. The proposed regulation network is illustrated in Fig 3.

**Hypoxia-induced DNA replication stress in gilthead seabream hepatocytes is synergistically mediated by the hypoxia-inducible factor and mTORC1.** In the HYP trial, GSEA based on GO biological processes (BP) (Fig 4A and 4D), KEGG (Fig 4B and 4E), and REACTOME (Fig 4C and 4F) databases revealed a total of 249 significantly enriched (FDR < 0.05) terms (S6 Table).

Dissolved oxygen (DO) is one of the main limiting factors in fish farming as it can severely affect many aspects of fish performance and physiology. In ponds, it generally depends on phytoplankton's photosynthesis rate, aquatic organisms' respiration, and/or the atmospheric oxygen diffusion ($O_2$) [74]. De-oxygenation of the world's oceans has also recently been highlighted as a major consequence of climate change, which can impact offshore aquaculture [75]. $O_2$ is crucial in numerous cellular processes such as oxidative metabolism and energy supply through ATP generation. One of the many impairments caused by inefficient tissue oxygenation is genomic instability, which drives DNA replication stress. The negative enrichment of the pathways "DNA replication (ID: GO:0006260)", "DNA replication (ID: dre03030)", and "Cell cycle (ID: dre04110)" (Fig 4A–4C) indicates a potential stalling of DNA replication and a halt or a slowdown of the cell cycle in hypoxia-exposed fish. The "DNA Replication (ID: R-DRE-69306)" pathway was also found to be downregulated in metabolomics, as retrieved by the ORA (Fig 5B). DNA replication is the process of genome duplication that a cell undergoes during cell cycle division. In eukaryotes, it is initiated by the binding of the origin recognition complex (ORC) to a replication origin, which then recruits a hexameric DNA helicase (MCM) and a helicase loading factor to form the pre-replicative protein complex (pre-RC) [76]. Downregulation of the REACTOME pathway "Activation of the pre-replicative complex (ID: R-DRE-68962)" (Fig 4C) indicates a potential hypoxia-induced replication arrest due to impaired pre-RC assembly/activation. Several studies in humans have demonstrated that a decrease in dNTP levels accompanies abrogated replication under hypoxic conditions due to downregulation of ribonucleotide reductase (RNR), a key enzyme that mediates the synthesis of deoxyribonucleotides, the key blocks for DNA replication and repair [77,78]. In accordance with these findings, the gene coding for this enzyme, *rrm1*, was significantly downregulated in hypoxia-exposed fish (LFC = -1.77, padj = 0.008). Moreover, metabolites UMP, uracil and uridine, involved in nucleotide biosynthesis, were significantly downregulated in the liver of these fish, according to a metabolomics analysis [49]. This is in accordance with the downregulation of the KEGG pathway "Pyrimidine metabolism (ID: dre00240)" in both the transcriptome and metabolome (Fig 5B). Previous studies are in accordance with these findings, as nucleotide biosynthetic processes were also downregulated in the gills of golden Pompano (*Trachinotus ovatus*) under hypoxic stress [79]. Moreover, *rrm1* was also found to be downregulated in the gills of juvenile Chinook salmon (*Oncorhynchus tshawytscha*) under

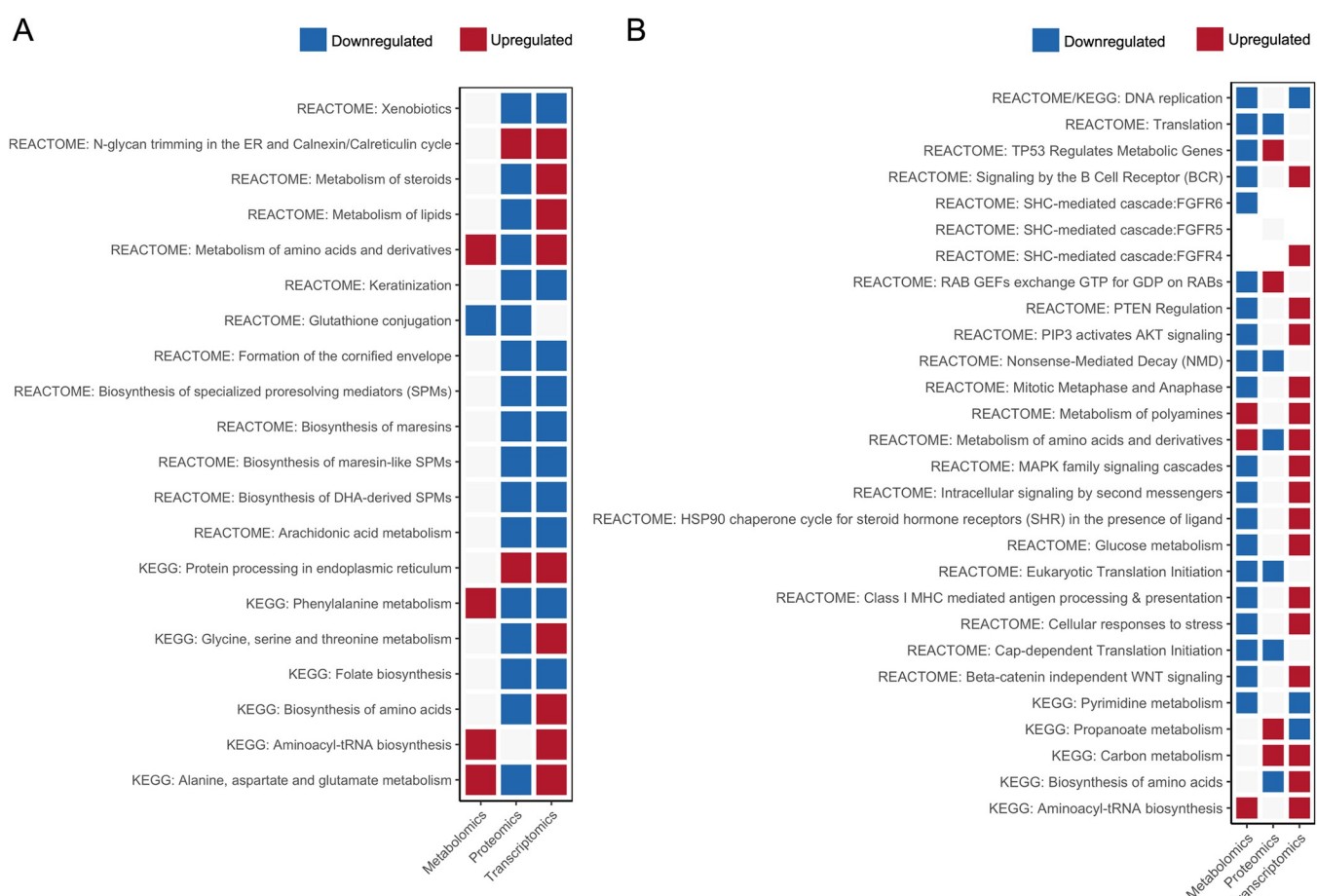

**Fig 4. GSEA of the liver RNA-seq data of gilthead seabream submitted to hypoxia.** Analysis was based on GO (A,D), KEGG (B,E), and REACTOME (C,F) databases, sorted by normalized enrichment score (NES) inferred from permutations of the gene set and false discovery rate (FDR). On the x-axis, the genes were ranked from the most upregulated (left end) to the most downregulated (right end). The y-axis represents the running enrichment score (ES). First line indicates downregulated pathways whereas bottom line indicates upregulated pathways.

hypoxic conditions for six days [80]. Hypoxia is also known to induce replication stress (RS) and activate the DNA damage response (DDR) independently of the DNA damage itself. This response relies on surveillance sensor kinases, namely the ataxia-telangiectasia-mutated kinase (ATM), ataxia telangiectasia and Rad3-related protein (ATR), and DNA-dependent protein kinase (DNA-PK), which are activated via PTMs. The activation of these pathways can result in the regulation of DNA repair pathways, cell cycle control, and apoptosis. Depending on the severity of hypoxia, that is, duration and level of oxygen, DNA repair pathways can be activated or repressed at the transcriptional level [77,78,81]. Negative enrichment of the pathways "double-strand break repair via homologous recombination (ID: GO:0000724)", "Activation of ATR in response to replication stress (ID: R-DRE-176187)", and "HDR through Single Strand Annealing (SSA) (ID: R-DRE-5685938)" (Fig 4A–4C; S6 Table), parallel with the significant upregulation of genes (e.g., *xrcc5*, *xrcc6*) involved in the canonical non-homologous end-joining repair mechanism (S3 Table), suggests a potential selective regulation of the DNA repair pathways, favouring the downregulation of some and the upregulation of others.

The relationship between hypoxia, cell cycle arrest, and DNA repair mechanism inhibition has not yet been completely revealed in teleosts. Nevertheless, downregulation of DNA replication due to hypoxia has also been observed in the gills of spotted seabass (*Lateolabrax*

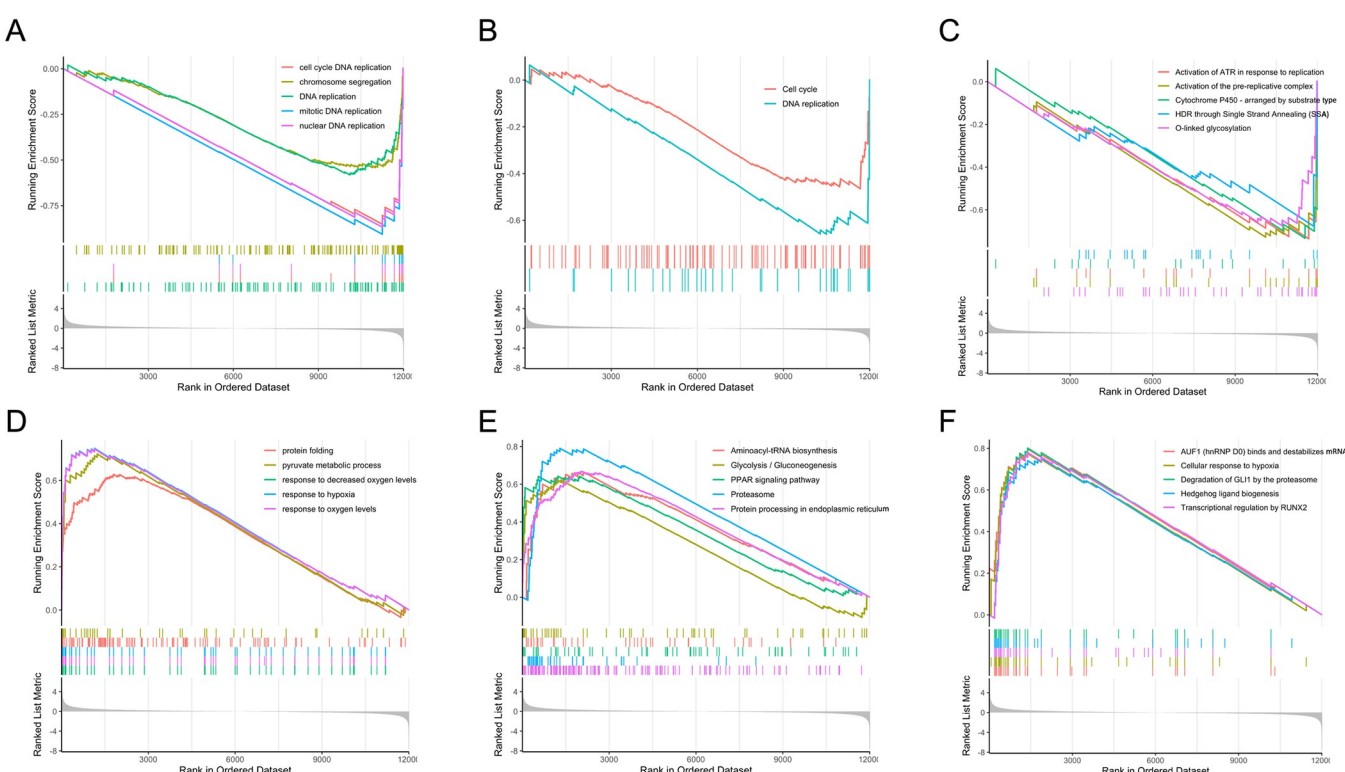

**Fig 5. Heatmap of multiomics overrepresentation analysis (ORA).** (A) net handling trial, (B) hypoxia trial; listed terms are commonly overrepresented terms between omics datasets.

*maculatus*) [82] and liver of threespine stickleback (*Gasterosteus aculeatus*) [83]. In Nile tilapia (*Oreochromis niloticus*), short and prolonged hypoxia induced DNA damage that was directly proportional to increasing hypoxic concentrations [84].

Hypoxia-inducible factor 1 (HIF-1) is at the centre of almost all hypoxia-induced pathways, acting mainly as a TF to mediate adaptive responses at both cellular and systemic levels. The isoform HIF-1α is a well-documented key modulator of the hypoxia signalling pathway; after being translocated into the nucleus, it heterodimerizes with HIF-1β and binds to hypoxia-responsive elements (HREs) located in the promoters of hypoxia-inducible genes, which modulate their expression [85]. In this study, several genes involved in the HIF-1 signalling pathway were significantly upregulated, including *egln1, egln2, egln3, hif1an, and hif1al* (S3 Table). Accordingly, the positive enrichment of the pathways "response to hypoxia (ID: GO:0001666)", "Cellular response to hypoxia (ID: R-DRE-1234174)" and "Oxygen-dependent proline hydroxylation of Hypoxia-inducible Factor Alpha (ID: R-DRE-1234176)" (Fig 4D–4F, S6 Table) further supports the activation of HIF-1α in the liver of hypoxia-exposed gilthead seabream. Furthermore, several DEGs known to be targeted by HIF-1α and to promote hypoxia adaptation through different mechanisms were also upregulated (S3 Table), such as *vegfaa*, which initiates the vascular endothelial growth factor (VEGF) signalling pathway [86]; *epoa*, which stimulates blood cell production, *higd1a* which is responsible for maintaining mitochondrial homeostasis, *slc2a1b* which facilitates cellular glucose uptake [87], *gapdhs, pgk1, eno1a, slc16a3* and *ldha1*, metabolic enzymes that reduce oxygen consumption and promote anaerobic metabolism [88], *pdk1* which inhibits the tricarboxylic acid (TCA) cycle [89] and *bnip3lb* which promotes mitophagy [88]. These changes can be supported by the positive enrichment of the pathways "Glycolysis/Gluconeogenesis (ID: dre00010)", "Mitophagy–

animal (ID: dre04137)" and "Glycolysis (ID: R-DRE-70171)" (Fig 4E; S6 Table). In addition to metabolism, HIF-1α has also been demonstrated to inhibit the activation of the MCM helicase in a non-transcriptional manner [90], which might corroborate the downregulation of DNA replication initiation, as genes *mcm2*, *mcm3*, *mcm4*, and *mcm5* were significantly downregulated in hypoxia-exposed fish (S3 Table). Additionally, cell cycle arrest can also be induced in a HIF-1α-dependent manner by displacing c-Myc from the p21 and p27 promoters, two cyclin-dependent kinases (CDKs) inhibitors [91]. Here, the genes *myca* (LFC = -0.91, padj = 0.006) and *cdkn1a* (LFC = 1.28, padj = 0.004), encoding the proteins c-Myc and p21, respectively, were downregulated and upregulated in response to hypoxia, supporting the action of HIF-1α in cell cycle arrest.

The relationship between hypoxia-induced activation of HIF-1 and metabolism has also been widely demonstrated in the livers of different fish species exposed to hypoxic conditions, such as *Epinephelus coioides* [92], *Procambarus clarkia* [93], *Hypophthalmichthys nobilis* [94], *Salvelinus alpinus* [95] and *Danio rerio* [96]. Curiously, *hif-1α* was downregulated in the white skeletal muscle and the heart of gilthead seabream subjected to moderate hypoxia (42–43%) [97], suggesting that either only more severe hypoxic conditions, such as the 15% oxygen saturation applied in this study, are able to induce HIF-1α activation in this species, or that the response of this factor differs significantly among tissues.

Another major signalling pathway that responds to hypoxia and promotes adaptation to low $O_2$ availability is mTORC1. In another study, mTORC1 signalling has been reported to be downregulated in Arctic char exposed to 15% DO [95]. Hypoxic conditions are known to lead to a downregulation of OXPHOS and, thus, to a reduction in cellular energy, consequently ceasing high-energy-demanding cellular processes such as translation. A metabolomic analysis of the livers of the same fish confirmed that ATP levels were significantly downregulated [49]. This can lead to an inhibition of the mTORC1, mediated by the metabolic regulator 5' AMP-activated protein kinase (AMPK) and/or the Regulated in DNA damage and development 1 (REDD1) [98]. The latter activates TSC2 by titrating the inhibitory 14-3-3 proteins [99]. In this study, *ddit4* (LFC = 2.81, padj = 2.20e-10) and *ywhaz* (LFC = 0.89, padj = 0.0007), coding for the proteins REDD1 and 14-3-3, respectively, were significantly upregulated in hypoxia-exposed fish, suggesting that REDD1 might be important for maintaining cellular energy homeostasis during oxygen challenges in this species. Gene *ddit4* was likewise found to be upregulated in threespine stickleback (*Gasterosteus aculeatus*) [83], in the gills and heart of bighead carp (*Hypophthalmichthys nobilis*) [94], and in the muscle of largemouth bass (*Micropterus salmoides*) [100], exposed to different hypoxia levels. To promote autophagic cell death mTORC1 can also be inhibited by BNIP3, which is transcriptionally activated by HIF-1α [101]. As previously mentioned, *bnip3lb* was significantly upregulated in hypoxia-exposed fish (LFC = 1.72, padj = 8.48e-08), demonstrating an inhibitory effect of HIF-1α over mTORC1. A transcriptomic analysis of zebrafish exposed to hypoxia revealed increased levels of *bnip3* in the heart [102]. Similarly, *bnip3* was also upregulated in channel catfish infected with *Edwardsiella ictaluri* [103]. Finally, mTORC1 could also be inhibited by a downregulation of the IGF1 signalling, as *igfbp1a* was also upregulated in response to hypoxia (LFC = 4.20, padj = 2.22e-10), as observed in net-handled fish. Previously *in vivo* and *in vitro* studies with zebrafish embryos demonstrated unequivocal evidence of a causal relationship between elevated IGFBP1 expression and hypoxia-induced embryonic growth and developmental retardation, suggesting that the HIF pathway is responsible for its transcriptional activation [104]. Another study reported that the zebrafish IGFBP-1 promoter contains 13 consensus hypoxia response elements (HREs) [105]. This protein was also upregulated at the mRNA level in Atlantic croaker during hypoxic stress [106]. The proposed regulation network is illustrated in Fig 3.

**The dual role of the endoplasmic reticulum in the adaptation to net handling and hypoxia stress: Cholesterol biosynthesis and the unfolded protein response.** Regarding the upregulated pathways in net-handled fish, one of the most enriched in both the GSEA and ORA was steroid and cholesterol biosynthesis (ID: GO:0006695, dre00100, R-DRE-191273), which takes place in the endoplasmic reticulum (ER) (Fig 2D–2F). Cholesterol, which can be obtained from diet or synthesized *de novo* in the liver, is a crucial component of the cell membranes in vertebrates and a precursor of the stress hormone cortisol [107]. Sterol responsive element binding protein (SBREBP) is the master transcriptional regulator of cholesterol biosynthesis, mediated by mTORC1 [108]. *Srebf2*, the gene encoding for this protein, was found to be significantly upregulated in net-handled fish (LFC = 1.18, padj = 0.0001), together with multiple genes involved in steroid and cortisol synthesis and in pathways downstream of cortisol action (*cyp21a2, hsd17b7, fdft1, stard, cebpb, pck1, pck2, g6pca.1*) (S3 Table). On the other hand, in hypoxia-exposed fish, multiple DEGs involved in steroid and cholesterol biosynthesis were downregulated (*cyp7b1, hsd17b7, hsd17b1, cyp2r1, and fdft1*), which could be explained by the likely downregulation of mTORC1, as previously hypothesized. In addition, the plasma cortisol levels of these fish were found to be significantly upregulated in NET fish and unchanged in HYP fish [27]. Cortisol, considered the primary stress hormone in fish, is a multifaceted glucocorticoid synthesized by interrenal cells in the head kidney as a quick response to external stimuli. It is vastly studied in teleost fish and is widely used as a physiological stress marker [109]. However, there is still a lack of studies on the association between cholesterol biosynthesis and cortisol response in stressed fish.

The pathways "Protein processing in endoplasmic reticulum (ID: dre04141)", "response to endoplasmic reticulum stress (ID: GO:0034976)" and "Asparagine N-linked glycosylation (ID: R-DRE-446203)" (Fig 2A–2C) were also positively enriched in net-handled fish, suggesting that this challenge might have induced stress in the ER, which is in accordance with the ORA of the proteomics data (Fig 5A). In fact, several processes related to ER stress were modulated by the challenge, specifically the N-glycan trimming, the ER quality control, the ER-associated degradation (ERAD), the ubiquitin ligase complex and the unfolded protein response (UPR). Associated to these pathways, 11 genes were significantly upregulated (LFC > 1, padj < 0.01), i.e., *calr, calr3b, ddost, canx, prkcsh, dnajb11, sar1ab, hyou1, pdia6,* and *pdia4* (S3 Table). In addition to being the organelle responsible for lipid synthesis and protein folding, the ER is the most important storage site for intracellular calcium ions. The newly synthesized proteins are translocated into the ER lumen and glycosylated. Correctly folded proteins are then transported to the Golgi complex, while misfolded proteins are targeted by chaperones for refolding or degradation through the ER-associated degradation (ERAD) system if terminally misfolded [110]. When homeostasis is compromised by conditions such as hypoxia, nutrient deprivation, calcium depletion, or accumulation of misfolded proteins, stress is induced, which initiates the unfolded protein response (UPR). Three ER-transmembrane stress sensors mediate this signal transduction pathway: inositol-requiring enzyme 1 (IRE1), pancreatic endoplasmic reticulum kinase (PERK), and activating transcription factor 6 (ATF6). The three branches of the UPR converge to restore homeostatic adaptation; however, in severe cases, they can switch to promote apoptotic cell death [111]. One of the main outputs of PERK signalling is the attenuation of translation through the inhibitory action of EIF4EBP1 (*eif4ebp1*, LFC = -0.47, padj = 0.0099). Specifically, cap-dependent translation is temporarily downregulated, in tandem with increased cap-independent translation of many mRNAs, such as activating transcription factor 4 (ATF4) [112]. The ATF6 and IRE1 pathways regulate the expression of genes mainly involved in protein folding and ERAD, which were significantly upregulated by the challenge (e.g., *calr, pdia6, dnajb11,* and *hyou1*). The results suggest that fish exposed to net

handling likely counteracted ER stress by activating the UPR and ERAD and avoiding cell death (Fig 3).

Similarly to net-handled fish, "Protein processing in the endoplasmic reticulum (ID: dre04141)" was significantly enriched in hypoxia exposed fish and among the top 5 pathways with the highest NES, along with "Asparagine N-linked glycosylation (ID: R-DRE-446203)", "protein folding (ID: GO:0006457)" and "response to unfolded protein (ID: GO:0006986)" (Fig 4D–4F; S6 Table). Hypoxia can induce protein misfolding due to the lack of oxygen required to form disulphide linkages, leading to ER stress and the consequent activation of the UPR. In this study, several DEGs were involved in distinct processes in the ER, such as *vcp*, *prkcsh*, *uggt1*, *plaa*, *hspa5*, *pdia6*, *calr*, *hsp90aa1.2*, *ero1a*, *hsp70.3*, and *xbp1*. Additionally, seven DEGs encoding proteasome subunits were also significantly upregulated, coupled with the positive enrichment of the pathways "ERAD pathway (ID: GO:0036503)" and "Proteasome (ID: dre03050)" by GSEA. These results suggest a hypoxia-mediated response of the ER, based on the activation of the UPR and ERAD pathways, to deal with misfolded proteins and maintain ER homeostasis. In a study using DNA microarrays, UPR was also upregulated in the liver of gilthead seabream exposed to low temperatures [113]. Also in gilthead seabream, genes involved in lectin chaperone-mediated protein quality control were found to be upregulated in response to mild hypoxia [7]. In rainbow trout subjected to heat stress, an RNA-seq study also revealed upregulation of the KEGG pathway "Protein processing in the ER" [15].

In the case of unresolved and/or sustained ER stress, the kinase domain of IRE1 has been shown to activate the Jun N-terminal kinase (JNK) signalling pathway, which apart from being implicated in ER stress-related apoptosis it also promotes cell survival by inducing autophagy [114]. The transcription factor Jun is a central JNK target in the promotion of hepatocyte survival and in this study *junba* and *junbb* encoding this protein were significantly upregulated in fish exposed to both hypoxia (*junba*: LFC = 1.00, padj = 0.006; *junbb*: LFC = 0.84, padj = 0.02) and net handling (*junba*: LFC = 1.79, padj = 3.48e-08; *junbb*: LFC = 0.73, padj = 0.049) challenges, suggesting an important role in stress adaptation in gilthead seabream that requires further investigation. The pathway "autophaghy (ID: GO:0006914)" was also positively enriched in hypoxia-exposed fish, which is concomitant with the downregulation of the mTORC1 pathway. *In vitro* and *in vivo* in mice, have demonstrated that Jun protected the hepatocytes from excessive activation of the ER stress response and subsequent cell death, linking the UPR to autophagy [115]. Jun was also significantly upregulated in the liver of rainbow trout exposed to confinement stress [116] and handling [68].

## Conclusion

Altogether, the results showed a challenge-specific transcriptional response of the liver of gilthead seabream to the different stimuli imposed, reinforcing the high phenotypic plasticity of this species to the changing environment. The most pronounced difference was observed between the overcrowded fish, and the fish exposed to net handling and hypoxia challenges, in terms of the number of dysregulated genes and gene families. Gilthead seabream has demonstrated high resilience to high stocking densities (45 kg m$^{-3}$), which might be due to domestication and/or evolutionary adaption, in contrast to what was observed in fish netted four times a week and exposed to 15% DO. Net-handled and hypoxia-exposed fish also demonstrated specific responses, such as the ribosome assembly stress response and DNA replication stress, respectively; however, both appeared to converge in the attenuation of translation to avoid proteotoxicity and shift the energy from cell proliferation and somatic growth towards stress-coping pathways. Notwithstanding, the response to both stressors converged in the induction of ER stress and downregulation of insulin growth factor signalling, a pathway that regulates

many of the downstream processes described here. It is also important to note that a complete understanding of these responses was only made possible by the integration of biological data from the different complementary molecular levels, showing the promisor role of multiomics in understanding the fate of mRNA and the complete picture of the stress response pathways.

The characterization and identification of potentially novel genes represents the next step towards a more holistic understanding of the coping mechanisms to stressful aquaculture routines. Within this framework, knowledge of the genetic background of commercially important fish species that efficiently adapt to challenging conditions can provide evidence of desirable traits that can be a win-win strategy for overcoming both animal welfare and sustainability issues in aquaculture.

## Supporting information

**S1 Fig. Methodology workflow.** Schematic workflow of the experimental trials and transcriptomics analysis.
(TIF)

**S1 Table. Paired-end reads per sample.** Number of paired-end reads ($2 \times 151$ bp) obtained by RNA-seq (Illumina NovaSeq 6000 System, with poly-A selection) from the 54 liver samples of gilthead seabream (*Sparus aurata*) submitted to overcrowding (OC), net handling (NET), and hypoxia (HYP).
(XLSX)

**S2 Table. Normalized counts (DESeq2) of assembled genes identified in the liver (n = 9) of gilthead seabream subjected to overcrowding (OC), net handling (NET), and hypoxia (HYP).** A reference-guided transcriptome assembly was carried out using Stringtie, followed by mapping using STAR aligner. The corresponding zebrafish (*Danio rerio*) ortholog is given for each gene.
(XLSX)

**S3 Table. Differentially expressed genes, retrieved by Wald's test within DESeq2, in the livers (n = 9) of gilthead seabream subjected to overcrowding (OC), net handling (NET), and hypoxia (HYP).** Differences were considered statistically significant when adjusted p-value (Benjamini-Hochberg correction) < 0.01 and log2|fold-change| (LFC) > 1.0. Corresponding zebrafish (*Danio rerio*) ortholog is given for each gene.
(XLSX)

**S4 Table. Gene set enrichment analysis of the genes identified in the liver of gilthead seabream subjected to net handling.** The analysis was based on GO, KEGG and REACTOME knowledgebases.
(XLSX)

**S5 Table. Multiomics overrepresentation analysis of differential genes, proteins and metabolites identified in the liver of gilthead seabream subjected to net handling (NET) and hypoxia (HYP).** Proteomics and metabolomics datasets were retrieved from Raposo de Magalhães et al. 2022 (https://doi.org/10.3390/ijms232315395).
(XLSX)

**S6 Table. Gene set enrichment analysis of the genes identified in the liver of gilthead seabream subjected to hypoxia.** The analysis was based on GO, KEGG and REACTOME knowledgebases.
(XLSX)

**S1 File. Multi-QC report integrating the results from the RNA-seq data pre-processing and analysis.**
(HTML)

## Acknowledgments

The authors would like to thank Dr. Rita Teodósio from CCMAR for advising on RNA isolation.

## Author Contributions

**Conceptualization:** Cláudia Raposo de Magalhães, Marco Cerqueira, Pedro M. Rodrigues.

**Data curation:** Cláudia Raposo de Magalhães, Ferenc Kagan.

**Formal analysis:** Cláudia Raposo de Magalhães, Kenneth Sandoval, Grace McCormack, Ana Paula Farinha.

**Funding acquisition:** Pedro M. Rodrigues.

**Investigation:** Cláudia Raposo de Magalhães, Grace McCormack, Denise Schrama, Raquel Carrilho, Marco Cerqueira.

**Methodology:** Cláudia Raposo de Magalhães, Kenneth Sandoval, Denise Schrama, Raquel Carrilho, Pedro M. Rodrigues.

**Project administration:** Pedro M. Rodrigues.

**Resources:** Pedro M. Rodrigues.

**Software:** Ferenc Kagan.

**Supervision:** Marco Cerqueira, Pedro M. Rodrigues.

**Validation:** Cláudia Raposo de Magalhães, Pedro M. Rodrigues.

**Visualization:** Cláudia Raposo de Magalhães.

**Writing – original draft:** Cláudia Raposo de Magalhães.

**Writing – review & editing:** Cláudia Raposo de Magalhães, Kenneth Sandoval, Ferenc Kagan, Grace McCormack, Denise Schrama, Raquel Carrilho, Ana Paula Farinha, Marco Cerqueira, Pedro M. Rodrigues.

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
