## [Decision Letter · Decision Letter 0]

20 Dec 2023

PONE-D-23-23892Transcriptomic insights into Sparus aurata hepatic response to different aquaculture challenges: A comprehensive RNA-seq and multiomics integration studyPLOS ONE

Dear Dr. Leal Rodrigues,

Thank you for submitting your manuscript to PLOS ONE. After careful consideration, we feel that it has merit but does not fully meet PLOS ONE’s publication criteria as it currently stands. Therefore, we invite you to submit a revised version of the manuscript that addresses the points raised during the review process.

We look forward to receiving your revised manuscript.

Kind regards,

A. K. Shakur Ahammad, PhD

Academic Editor

PLOS ONE

“This work is integrated into the project WELFISH (Refª 16-02-05-FMP-12) financed by Mar2020, in the framework of Portugal 2020. Additionally, it received Portuguese national funds from FCT - Foundation for Science and Technology through projects UIDB/04326/2020, UIDP/04326/2020, and LA/P/0101/2020, from the operational programs CRESC Algarve 2020 and COMPETE 2020 through projects EMBRC.PT ALG-01-0145-FEDER-022121 and BIODATA.PT ALG-01-0145-FEDER-022231, and from the European Union’s Horizon 2020 research and innovation program under grant agreement No 730984, ASSEMBLE Plus project, and Marie Skłodowska-Curie grant agreement No. 764840. Cláudia Raposo de Magalhães acknowledges a FCT PhD scholarship, Refª SFRH/BD/138884/2018, funded by national funds from MCTES, via FCT, and co-funded by the European Union through the European Social Fund (ESF). Denise Schrama acknowledges a FCT PhD scholarship, Refª SFRH/BD/136319/2018. Raquel Carrilho acknowledges a FCT PhD scholarship, Refª 2021.06786.BD. Ana Paula Farinha acknowledges postdoctoral fellowship, Refº MAR-02.05.01-FEAMP-0012 (MAR2020). Marco Cerqueira acknowledges a FCT contract, Refª 2020.02937.CEECIND.”

Please respond by return e-mail so that we can amend your financial disclosure and competing interests on your behalf.

4. We note that Figure 3 in your submission contain copyrighted images. All PLOS content is published under the Creative Commons Attribution License (CC BY 4.0), which means that the manuscript, images, and Supporting Information files will be freely available online, and any third party is permitted to access, download, copy, distribute, and use these materials in any way, even commercially, with proper attribution. For more information, see our copyright guidelines: http://journals.plos.org/plosone/s/licenses-and-copyright.

1. You may seek permission from the original copyright holder of Figure 3 to publish the content specifically under the CC BY 4.0 license.

Reviewers' comments:

Reviewer's Responses to Questions

**Comments to the Author**

1. Is the manuscript technically sound, and do the data support the conclusions?

Reviewer #1: Partly

Reviewer #2: Yes

2. Has the statistical analysis been performed appropriately and rigorously? 

Reviewer #1: Yes

Reviewer #2: Yes

3. Have the authors made all data underlying the findings in their manuscript fully available?

Reviewer #1: Yes

Reviewer #2: Yes

4. Is the manuscript presented in an intelligible fashion and written in standard English?

Reviewer #1: Yes

Reviewer #2: Yes

5. Review Comments to the Author

Reviewer #1: The article is relevant to the field of aquaculture and fish farming, as it assesses areas that still need exploration. The work related to the transcriptome presents an adequate methodology. However, there is room for improvement in some aspects regarding the language of transcriptomics, proteomics, and metabolomics.

Approach:

As I began reading, I found the discussion of different omics aspects intriguing. However, when delving into the methodology to understand the techniques, analyses, and potential combinations of omics techniques, I realized that the analysis carried out focused solely on the transcriptome. The various analyses provided in the supplementary material are informative, yet I did not observe any simple or more complex analyses of proteomics techniques such as 2DGE, Ion-Trapping ESI-MS, MALDI-MS (mass spectrum), and others.

Instead, it would be beneficial to extrapolate aspects of transcriptome analysis to encompass proteins and metabolisms. By doing so, we can provide information and targets for other omics. This approach will guide future work, comparisons, and hypotheses about liver function. However, it's important to note that we cannot derive all aspects of other omics solely from transcriptomics data.

Recent works in Proteomics have even discussed the complexity in a Multi-Omics world. Other studies have examined the RNA-seq approach in relation to mass spectrometry techniques, revealing that gene expression data can serve as an indirect reference for protein level comparisons. Nevertheless, in experiments involving animals and humans, it's clear that using only transcript data is insufficient. For instance, Reactome is a free online database of biological pathways, but despite data integration, it includes in-silico analysis deposits, leading to compromised protein information without in-vitro or in-vivo validation.

Hence, I strongly recommend using transcriptomics to raise points for other omics within the obtained data. Despite its versatility, transcriptomics has limitations in making reliable statements about other omics.

Regarding transcript analysis, I suggest clarifying the genomes used for mapping, as this will impact future comparisons. Clearly state the name of the deposited sequence used.

I recommend publication with some approach modifications.

Supplementary material:

To enhance accessibility for those unfamiliar with transcriptomic data, consider including a legend with a gradual color scale of the transcript data (S1, S3, and S4).

If you wish to replicate the methodology approach, creating a figure depicting the experimental design (with the 54 samples obtained) and placing it in the supplementary material would be beneficial.

Questions and other suggestions:

The figures appear to have low resolution, potentially due to the submission system. I recommend emailing the figures directly to address the quality issue.

Specify the genome or genomes used for assembly, including the deposit name and bank. Even if this information is in the supplementary material.

Provide details about the number of fish used. Since there may be variations among species, this information would be valuable for readers interested in comparisons.

How long did the adult fish arrive with the company 'Maresa, Seafood from Estero S.A.' (Huelva, Spain)? What is the average lifespan?

In Figure 1, include a caption explaining the significance of each lettered item (A, B, C, D, E, and F).

I also noticed the absence of Figure 3, which could provide a well-executed representation of down or up regulation. Including relevant regulation variations would enhance its value.

When citing programs or packages, always specify the version used due to the potential for rapid changes.

Is there a reason for not discussing other aspects mentioned in the article (https://doi.org/10.3390/ijms232315395)?

Reviewer #2: Good research work, the topic was reviewed well and in detail, and the results were collected and interpreted accurately.The work was more accurate when the work was at the genetic RNA level. And it was doneTotal RNA extraction and purification Total RNA was extracted

6. PLOS authors have the option to publish the peer review history of their article (what does this mean?). If published, this will include your full peer review and any attached files.

Reviewer #1: No

Reviewer #2: No

---

## [Author Response · Author response to Decision Letter 0]

13 Jan 2024

Editor’s comments:

Style corrections were performed according to the templates.

Modifications were perfomed accordingly.

“This work is integrated into the project WELFISH (Refª 16-02-05-FMP-12) financed by Mar2020, in the framework of Portugal 2020. Additionally, it received Portuguese national funds from FCT - Foundation for Science and Technology through projects UIDB/04326/2020, UIDP/04326/2020, and LA/P/0101/2020, from the operational programs CRESC Algarve 2020 and COMPETE 2020 through projects EMBRC.PT ALG-01-0145-FEDER-022121 and BIODATA.PT ALG-01-0145-FEDER-022231, and from the European Union’s Horizon 2020 research and innovation program under grant agreement No 730984, ASSEMBLE Plus project, and Marie Skłodowska-Curie grant agreement No. 764840. Cláudia Raposo de Magalhães acknowledges a FCT PhD scholarship, Refª SFRH/BD/138884/2018, funded by national funds from MCTES, via FCT, and co-funded by the European Union through the European Social Fund (ESF). Denise Schrama acknowledges a FCT PhD scholarship, Refª SFRH/BD/136319/2018. Raquel Carrilho acknowledges a FCT PhD scholarship, Refª 2021.06786.BD. Ana Paula Farinha acknowledges postdoctoral fellowship, Refº MAR-02.05.01-FEAMP-0012 (MAR2020). Marco Cerqueira acknowledges a FCT contract, Refª 2020.02937.CEECIND.”

Please state what role the funders took in the study. If the funders had no role, please state: "The funders had no role in study design, data collection and analysis, decision to publish, or preparation of the manuscript.

Please respond by return e-mail so that we can amend your financial disclosure and competing interests on your behalf.

The required statement was added to the financial disclosure.

4. We note that Figure 3 in your submission contain copyrighted images. All PLOS content is published under the Creative Commons Attribution License (CC BY 4.0), which means that the manuscript, images, and Supporting Information files will be freely available online, and any third party is permitted to access, download, copy, distribute, and use these materials in any way, even commercially, with proper attribution. For more information, see our copyright guidelines: http://journals.plos.org/plosone/s/licenses-and-copyright.

Figure 3 does not contain copyrighted images. The figure was entirely created using shapes from PowerPoint and Inkscape.

Reviewer #1: The article is relevant to the field of aquaculture and fish farming, as it assesses areas that still need exploration. The work related to the transcriptome presents an adequate methodology. However, there is room for improvement in some aspects regarding the language of transcriptomics, proteomics, and metabolomics.

All authors are deeply thankful for the opportunity to submit a revised draft of this manuscript for publication in the PLOS ONE Journal. We appreciate the time and effort that the reviewers dedicated to providing feedback on our manuscript and are grateful for the constructive comments and suggestions that truly improved our paper. We have incorporated most of the suggestions made by the reviewers. Please see below, in blue, for a point-by-point response to the reviewers’ comments.

Approach:

As I began reading, I found the discussion of different omics aspects intriguing. However, when delving into the methodology to understand the techniques, analyses, and potential combinations of omics techniques, I realized that the analysis carried out focused solely on the transcriptome. The various analyses provided in the supplementary material are informative, yet I did not observe any simple or more complex analyses of proteomics techniques such as 2DGE, Ion-Trapping ESI-MS, MALDI-MS (mass spectrum), and others.

Instead, it would be beneficial to extrapolate aspects of transcriptome analysis to encompass proteins and metabolisms. By doing so, we can provide information and targets for other omics. This approach will guide future work, comparisons, and hypotheses about liver function. However, it's important to note that we cannot derive all aspects of other omics solely from transcriptomics data.

Recent works in Proteomics have even discussed the complexity in a Multi-Omics world. Other studies have examined the RNA-seq approach in relation to mass spectrometry techniques, revealing that gene expression data can serve as an indirect reference for protein level comparisons. Nevertheless, in experiments involving animals and humans, it's clear that using only transcript data is insufficient. For instance, Reactome is a free online database of biological pathways, but despite data integration, it includes in-silico analysis deposits, leading to compromised protein information without in-vitro or in-vivo validation.

Hence, I strongly recommend using transcriptomics to raise points for other omics within the obtained data. Despite its versatility, transcriptomics has limitations in making reliable statements about other omics.

Thank you for your comment. Indeed analysis in this article was focused on transcriptomics, as proteomics and metabolomics has already been done in a previous work (https://doi.org/10.3390/ijms232315395). Shotgun label-free proteomics analysis is already a powerful and extremely sensitive high-throughput technique, thus we believe it wouldn’t make sense to perform simpler analyses. Regarding more complex analysis, such as TMT or iTRAQ, we believe that considering the main objective of the work, which was a stress proteome mapping, it was not necessary to perform a label-based quantification. Nevertheless, the techniques used in the other omics were added to the M&M section, to facilitate accessibility. Multiomics data can be analyzed by modelling omics datasets together or by integrating the findings of single omics datasets (e.g., integration of enriched biological functions). The first approach was what was done is our previous work, robust integrative frameworks and algorithms should be employed to avoid biased analysis towards a data modality with significantly more features (e.g., Data Integration Analysis for Biomarker discovery using Latent components (DIABLO), Multi-Omics Factor Analysis (MOFA), and joint and individual variation explained (JIVE)). The second approach is what was done in this manuscript. Additionally, weak correlations can often occur between different data modalities, particularly between proteomic and transcriptomic data. This is mainly due to the complex protein kinetics (e.g., PTMs) as to RNA splicing processes and variations in transcript half-lives and translation products. 

Regarding transcript analysis, I suggest clarifying the genomes used for mapping, as this will impact future comparisons. Clearly state the name of the deposited sequence used.

The genome used for mapping is already specified in lines 198-199, with a direct link to the assembly, as: “…Mapping to the Sparus aurata reference genome (Genome assembly: GCA_900880675.1, https://www.ensembl.org/Sparus_aurata/Info/Index)”

I recommend publication with some approach modifications.

Supplementary material:

To enhance accessibility for those unfamiliar with transcriptomic data, consider including a legend with a gradual color scale of the transcript data (S1, S3, and S4).

Thank you for your suggestion, although the color scale is already present in the supplementary materials mentioned, using a universal color scale commonly used in omics datasets (blue – downregulated, red – upregulated). Please let us know if the colors do not show up after downloading from PLOS one.

If you wish to replicate the methodology approach, creating a figure depicting the experimental design (with the 54 samples obtained) and placing it in the supplementary material would be beneficial.

 A figure representing the methodology workflow was created and included as “S1 Figure”.

Questions and other suggestions: The figures appear to have low resolution, potentially due to the submission system. I recommend emailing the figures directly to address the quality issue.

Thank you for the note. Pictures are all saved in 600 dpi but usually during the review process the journal does not provide the high-resolution versions, or sometimes these need to be downloaded separately from the PDF.

Specify the genome or genomes used for assembly, including the deposit name and bank. Even if this information is in the supplementary material.

The genome used for mapping is already specified in lines 198-199, with a direct link to the assembly, as: “…Mapping to the Sparus aurata reference genome (Genome assembly: GCA_900880675.1, https://www.ensembl.org/Sparus_aurata/Info/Index)”

Provide details about the number of fish used. Since there may be variations among species, this information would be valuable for readers interested in comparisons.

Thank you for your comment. Regarding the total number of fish used, this is indicated in line 140, in density unit (10 kg/m3), as the number of fish changed according to experimental condition tested, especially in the overcrowding challenge. Regarding the number of fish sampled, as indicated in line 160, the number of fish used was 9 fish per experimental group, corresponding to 3 fish per tank.

How long did the adult fish arrive with the company 'Maresa, Seafood from Estero S.A.' (Huelva, Spain)? What is the average lifespan?

Transport from the farm, with a certified company (Flying Sharks), to the experimental facilities took around 2h. Transports for around 8h are demonstrated not to affect gilthead seabream welfare. Fish acclimated for 2 weeks before starting the trials. Regarding the lifespan question, we did not exactly understand what the reviewer intended to know. 

In Figure 1, include a caption explaining the significance of each lettered item (A, B, C, D, E, and F).

In the figure 1 caption, which is in lines 286-292, the significance of the capital letters is already indicated, as follows: “Fig 1. Summary of the exploratory and differential analyses results of RNA-seq data. Biplots represent the principal component analyses (PCA) of the liver transcriptome of gilthead seabream submitted to overcrowding (A), net-handling (B), and hypoxia (C). Experimental groups are distinguished by different colours, as indicated in the legend. Arrows depict the top loadings. MA plots of the shrunken LFCs indicate differentially expressed genes: (D) overcrowding, (E) net handling, (F) hypoxia. Blue points represent padj > 0.01, and horizontal lines indicate the threshold of log2|fold-change| > 1.0.”

I also noticed the absence of Figure 3, which could provide a well-executed representation of down or up regulation. Including relevant regulation variations would enhance its value.

We are not completely sure about the reviewer’s comment. If the reviewer refers to an “absence” of Figure 3, we can confirm that this was submitted. Maybe due to some error in the PDF generator the figure was not somehow included. If the reviewer refers to the regulation of the pathways/genes, this is indicated in the figure caption in lines 386-388, as follows: “Fig 3. Proposed stress response network in gilthead seabream hepatocytes subjected to net handling and hypoxia. Dashed arrows indicate downregulated pathways, whereas solid arrows represent unchanged or upregulated pathways.”

When citing programs or packages, always specify the version used due to the potential for rapid changes.

We totally agree with the reviewer, although we believe that all software and R packages were mentioned together with the corresponding version. We appreciate if the reviewer can specify any that we might have missed.

Is there a reason for not discussing other aspects mentioned in the article (https://doi.org/10.3390/ijms232315395)?

The data integration, of the transcriptomics analysis with the proteomics and metabolomics data from the paper mentioned was done at the functional level. Thus, the pathways discussed were exclusively the ones in common between all omics datasets, as mentioned in M&M section.

Reviewer #2: Good research work, the topic was reviewed well and in detail, and the results were collected and interpreted accurately.The work was more accurate when the work was at the genetic RNA level. And it was done Total RNA extraction and purification Total RNA was extracted

All authors are deeply thankful for the opportunity to submit a revised draft of this manuscript for publication in the PLOS ONE Journal. We appreciate the time and effort that the reviewers dedicated to providing feedback on our manuscript and are grateful for the constructive comments and suggestions that truly improved our paper. We have incorporated most of the suggestions made by the reviewers. Please see below, in blue, for a point-by-point response to the reviewers’ comments.

---

## [Editor Report · Decision Letter 1]

27 Feb 2024

I hope this message finds you well. I would like to express my appreciation for your meticulous attention to the revisions requested by the reviewers and myself during the evaluation process of your manuscript entitled "Transcriptomic changes behind Sparus aurata hepatic response to different aquaculture challenges: an RNA-seq study and multiomics integration" submitted to PLOS ONE.I am pleased to inform you that the revisions author have made align with the constructive feedback provided by the reviewers. The significant revisions, as suggested by one reviewer, and accept, as suggested by the other, have collectively strengthened the manuscript, addressing both major and accept concerns. Having carefully reviewed the revised manuscript and considering the comprehensive revisions undertaken, I am confident that the manuscript is now ready for publication. The clarity of the content, adherence to journal guidelines, and the rigorous scientific approach demonstrated in authors work contribute to the high standards set by PLOS ONE.Therefore, I am delighted to convey that proposed manuscript may be accepted for publication in PLOS ONE. I believe manuscript will make a valuable contribution to the scientific community, and I am eager to see it published.Thank you once again for your diligence and dedication throughout the revision process. Please proceed with the final submission steps as guided by the journal's editorial office. Should you have any questions or require further assistance, please feel free to contact me.

---

## [Editor Report · Acceptance letter]

13 Mar 2024

PONE-D-23-23892R1 

PLOS ONE

Dear Dr. Rodrigues, 

I'm pleased to inform you that your manuscript has been deemed suitable for publication in PLOS ONE. Congratulations! Your manuscript is now being handed over to our production team.

Kind regards, 

on behalf of

Dr. A. K. Shakur Ahammad 

Academic Editor

PLOS ONE